# Lifestyles during the First Wave of COVID-19: A Cross-Sectional Study of 16,811 Adults from Spanish-Speaking Countries in South America

**DOI:** 10.3390/ijerph192215318

**Published:** 2022-11-19

**Authors:** Sergio Bravo-Cucci, Joan Chipia, Sandra Lobo, Lizmery López, Regina Munarriz-Medina, Tania Alvarado-Santiago, Rodrigo Núñez-Cortés

**Affiliations:** 1Academic Department of Medical Technology, Universidad Nacional Federico Villarreal, Lima 15007, Peru; 2School of Medicine, Los Andes University, Merida 5101, Venezuela; 3Academic Department of Administration, Universidad Nacional Federico Villarreal, Lima 15022, Peru; 4Medical Technology Research, Innovation and Entrepreneurship Unit, Universidad Nacional Federico Villarreal, Lima 15007, Peru; 5Department of Physical Therapy, Faculty of Medicine, University of Chile, Santiago 8240000, Chile

**Keywords:** lifestyles, COVID-19, social isolation, South America

## Abstract

The aim of this research was to analyze the lifestyles of adults from Spanish-speaking countries in South America during the COVID-19 pandemic using a cross-sectional, analytical, and multicenter study. The target population was composed of people aged 18 and older who resided in South America during the pandemic; the final sample comprised 16,811 participants who were predominantly female, with ages ranging from 18 to 79 years. The results showed that approximately six out of ten respondents did not engage in any physical activity; only one in four respondents indicated that their diet was sufficient and balanced; and most washed their hands frequently and showered every day. Regarding the type of isolation, half reported that it was voluntary and the rest mandatory or restricted. Regarding mobility, six out of ten surveyed leave their residence on a weekly basis. Regarding the use of personal protective equipment, the majority used masks and a smaller proportion used gloves. In conclusion, the majority of respondents did not perform any physical activity; moreover, one in four reported having a sufficient balanced diet. We therefore recommend an improvement of public policies to promote better lifestyles in South America, in particular the reorientation of the health system to prevent similar situations.

## 1. Introduction

In December 2019, pneumonia from an unknown cause appeared in China; authorities responded quickly, and in January 2020, the new coronavirus was isolated in Wuhan [1]. Based on the reports provided by China and other countries worldwide, the WHO declared this disease as a global health emergency at the end of January 2020 [2].

Most South American countries responded quickly to the new virus [3]. These measures became more rigorous and developed over time. This allowed for an adequate response and the control of illness progression [4]. Epidemiological surveillance and the early response were centralized through the health systems [5], which were waiting for a massive influx of vaccines to complement sanitary measures.

The SARS-CoV-2 pandemic has generated considerable changes in our current society. As mentioned earlier, several countries adopted different measures of social isolation that were either voluntary or compulsory. These measures involved changes in the lifestyles of families, communities, and populations [6]; among the changes generated by sanitary measures were changes in mobilization, which affected individuals’ diets and physical activity, which are considered favorable elements for health [7].

Mobilization restrictions can limit a population’s routine through the limitation of outdoor activities, including exercise or physical activity. The recommended daily level of activity is at least 30 min, in addition to high-intensity physical activity for at least 20 min some days [8]. During the pandemic, it must be recognized that outings and access to exercise spaces such as gymnasiums were restricted; thus, a reduction in physical activity was expected.

The combination of low physical activity and inappropriate diet causes an increased risk of obesity. This risk factor has been recognized as an accelerator of the pro-inflammatory factor, which causes a higher risk of complications, and thus, a worse outcome, with SARS-CoV-2 infection [8]. In one study, it was reported that confinement measures negatively influenced the health of individuals, for example by decreasing physical activity levels and worsening their diet; in addition, an increase in sedentary behavior was observed during quarantine periods [9].

A sedentary lifestyle is common in our current society, although its prevalence varies depending on age, especially after 60 years [10]. In this context, during the pandemic, an increase in non-transmissible illnesses, such as acute coronary syndrome, diabetes, some kinds of cancer, and even premature death [11], could be expected.

Another important risk factor for worse outcomes with COVID-19 is age [8]. However, it is important to emphasize that Latin America was better equipped to face this pandemic than European countries, which possess a considerable elderly population, a higher prevalence of noncommunicable diseases and different sociocultural conditions [12].

Lifestyles are the cause of the greatest burden of disease worldwide and in many regions such as Europe and South America, so we consider it important to investigate it in the context of the health management of a new disease such as COVID-19, where lifestyles could increase mortality from COVID-19 and the global burden of chronic diseases in the medium and long term. In addition, it was important to investigate the South American region early because knowing the lifestyles of its inhabitants would contribute to the literature and to explain future outcomes of the new pandemic [3,4]. The South American region was important to investigate early due to the health and cultural characteristics of its inhabitants could be important in the development of the new pandemic [3,4]. In addition, health inequalities caused by the social determinants of health have become clearly defined in this continent during the pandemic and could also be influencing lifestyles [13]. We aimed to analyze the lifestyles of adults such as hygiene, physical activity, mobilization, and diet of Spanish-speaking countries in South America during COVID-19.

## 2. Materials and Methods

For our quantitative research, we used an analytical, non-experimental, cross-sectional, and multicenter study design to collect and analyze information from several South American countries. The population of interest consisted of Spanish-speaking people older than 18 who were in social isolation due to COVID-19 in South America.

The final sample included individuals aged 18 years and over who resided in Spanish-speaking countries in South America (*n* = 16,811) during the COVID-19 pandemic.

The characteristics of the participants according to their country of residence and age group are shown in Table 1, where it is verified that Venezuela and Peru were the countries with the highest frequency. Regarding housing characterization, it was determined that 73% of the respondents lived in a house, 21.7% in an apartment, 4.6% in a neighborhood apartment, and 0.8% on a farm. Regarding dimensions, 65.5% indicated perceiving their housing size as normal, 24.0% as small, and 10.5% as large.

Our research variables included sociodemographic variables, the type of isolation, physical activity, diet, hygiene, outside mobilization, and housing characteristics.

A virtual survey was conducted using Google Form.

Instruments for data collection: A questionnaire was created containing items regarding sociodemographic variables, such as the type of social isolation, sex, age, residence country, nationality, instructional level, occupation, housing type, basic services, and housing conditions. Moreover, regarding lifestyles, the following areas were considered: Physical activity, diet, and personal and housing hygiene habits.

Data collection procedure: Before data collection, the protocol was submitted to the Bioethics Committee of Universidad de Los Andes (Los Andes University) through the Medicine School Council, who subsequently accepted it. Afterwards, the survey was carried out from 5 April 2020 to 26 April 2020.

Data Analysis: The indicators of frequency and percentage for the nominal qualitative variables were calculated descriptively, while the quantitative variables were calculated using the arithmetic mean, median, and standard deviation. Hypothesis tests using a 95% significance level were inferentially determined by applying the Chi-squared test in the cross of established variables and one-factor ANOVA. SPSS for Windows (version 24.0 Chicago, IL, USA) and Microsoft Excel 2019 were used to carry out the statistical analysis.

## 3. Results

The survey was completed by 17,447 Spanish-speaking South American people. Out of all the respondents, 16,811 were older than 18, consented to the use of their data and information, and completed the required data. The total sample was 87% (*n* = 14,622) female and 13% (*n* = 2189) male. The age range was from 18 to 79 years, with an average of 30.8 years, a median of 27 years, a typical deviation of 11.8 years, and a mean standard error of 0.091 years.

In relation to the number of rooms and people per house of the study participants. It can be seen that 1.6% of respondents lived in overcrowded conditions (more than three people per room). Most houses contained three rooms (32.6%), and most houses contained 4 people (24.6%).

Concerning the percentage of people who had access to basic services, it was determined that the highest percentage was for electricity at 98.7% (*n* = 16,458), followed by potable water at 97.9% (*n* = 16,593), internet at 84.3% (*n* = 14,171), and urban cleaning at 78.1% (*n* = 13,129).

With respect to physical activity, it was determined that 58.4% (*n* = 9815) of the respondents reported not doing any kind of physical activity, while 41.6% (*n* = 6996) did. Among the people who reported partaking in physical activities, it was found that 54.3%, 41.2%, and 41.5% reported doing moderate, low, and high physical activity, respectively. Regarding the weekly frequency of physical activity, it was found that the highest percentage reported was three times (28.4%), followed by two times (16.5%), five times (16.2%), four times (13.6%), seven times (9.2%), one time (8.2%), and six times (7.6%). Moreover, the calculated mean was 3.48 ± 1.785 times and the median was three times.

Concerning physical activity time (minutes), the average and standard deviation were calculated as 44.88 ± 28.851 min. With reference to the time respondents spent sitting or lying down, a mean and a typical deviation of 7.60 ± 5.229 h, a median of 6 h, and a mean standard error of 0.041 h were obtained. With regard to the most performed physical activity, exercise was first with 64.2% (*n* = 4491), followed by dancing with 28.6% (*n* = 2001), and other with 25.0% (*n* = 1749). Among the people who reported partaking in physical activity, 54.5% (*n* = 3813) felt pain; when asked about the location of recent pain, the highest percentage of pain was found in the neck (64.9%), followed by the spine or low back (60.2%), shoulders (42.4%), hips/legs (30.7%), knees (25.0%), hands/wrists (22.4%), ankles/feet (14.4%), and elbows (6.2%).

In relation to diet, the majority of respondents (61.7%) indicated that their diet was sufficient, 26% indicated it was sufficient and balanced, and 12.4% indicated it was insufficient. It was also shown that the most frequent meal was lunch (95.9%), followed by breakfast (80.6%), dinner (77.5%), and snacks (54.0%). In addition, it was noted that 60.2% (10,116) of the respondents did not eat during the recommended hours. In terms of the number of glasses of water consumed daily, 53.3% of the respondents indicated consuming fewer than five glasses, 33.6% from five to eight glasses, 9.2% more than eight glasses, and 3.9% no glasses.

Personal hygiene was also studied, and it was found that 73.8% of respondents reported taking a shower every day, while the rest (26.2%) indicated two times a week; moreover, 95.0% of respondents reported washing their hands often (after eating or performing an activity resulting in dirty hands), while 5% did not. Regarding house cleaning, 57.6% of respondents reported cleaning every day, 27.6% from two to four times a week, 12.3% once a week, and 2.5% once every two weeks; it was also found that the cleaning supplies that were most frequently used for house cleaning were bleach (82.5%) and soap (73.5%).

Concerning social isolation measures, 50.5% of respondents reported voluntary isolation, 42.6% compulsory isolation, and 6.9% restrictive isolation. Regarding mobilization, it was determined that 61.5% of the respondents left their residence; out of these individuals, 59.0% left once a week, 18.2% from two to four times a week, 12.8% every day, and 10.0% five or six times a week. With respect to the usage of personal protection equipment, 90.4% of respondents used masks and 31.7% used gloves.

The relations between hygiene, mobilization, and diet variables during quarantine among the study participants are shown in Table 2 and Figure 1.

Table 3 shows the significant differences between the type of isolation and physical activity (*p* = 0.021) and between the type of isolation and the intensity of the physical activity (*p* = 0.008). These differences were 95% reliable, according to the Chi-square test.

Table 4 shows the significant differences between means (Student’s *t*-test of independent samples, 95% reliable) when the time (hours) spent sitting or lying down was compared depending on social isolation modality (*p* < 0.001). It was found that the mean of respondents who indicated voluntary isolation was lower (7.28 h). In addition, the statistical relationship between the time (minutes) of physical activity and the type of isolation (*p* < 0.001) was determined; we concluded that the mean of individuals who indicated voluntary isolation was lower (42.13 min).

It should be noted that there were statistically significant differences between the number of times physical activity was performed and the type of isolation (*p* = 0.006), between physical activity and age range (*p* = 0.002), between physical activity and occupation (*p* < 0.001), and between the intensity of physical activity and occupation (*p* = 0.001), calculated via a Chi-square test. Moreover, the relationships between the intensity of physical activity and pain (*p* = 0.002) and between the number of times physical activity was performed and pain (*p* = 0.011) were found to be statistically significant.

Furthermore, regarding housing characterization, statistically significant differences were found between housing dimensions and diet quality (*p* < 0.001) and between housing type and diet quality (*p* < 0.001). Finally, significant differences were found between occupation and diet quality (*p* < 0.001) as well as between sex and diet (*p* < 0.001).

## 4. Discussion

The highest percentage of the voluntary respondents were women, people aged 30, and university students. These results were expected due to the fact that the female population tends to respond well to information exchange in a virtual way, contrary to the male population [14]. Moreover, it is known that the response level drops as individuals grow older, likely due to decreased access to the internet, lower income, or living alone [15]; on the contrary, individuals with tertiary education tend to respond more than those without.

The COVID-19 pandemic has negatively affected global health and caused significant changes in family, community, and population behaviors, particularly in terms of diet and physical activity [6], mainly due to social isolation and the economic consequences of the pandemic. The lifestyles of the studied population with respect to diet, physical activity, hygiene, and mobilization are described below.

The control of metabolic disorders has a positive impact on reducing the risk of developing severe forms of COVID-19 [16]. In addition, lifestyles associated with a bad diet (for instance, excessive consumption) can damage an adequate metabolic balance; this is also associated with a sedentary lifestyle and low levels of regular physical activity [16]. Long periods of social isolation may have impacts on common habits, which can be seen as, for example, an inadequate diet, an increase in sedentary patterns, and an increase in consumed calories [16]. This is in agreement with the results of this research, in which only 26% of the participants indicated having a sufficient and balanced diet during the quarantine period, a lower percentage than reported in Italy with 35.8% reporting eating healthy food, finding no significant changes before and after the pandemic [6], These low proportions of healthy eating could negatively influence how these populations coped whith COVID-19.

An Italian survey [6] found that only 13.4% of the population consumed >2 L of water a day; similarly, 9.2% of the participants in this study reported consuming the same amount.

Young adults tend to respond better to the incorporation of nutritional education to improve eating habits [17]; however, there is an association between eating disorders, young women, and teenagers [18]. This study did not find any association between age and the type of diet among our respondents, probably because the confinement measures forced many people to stay at home and prepare their food communally.

On the other hand, there was an association between sex and diet, corroborating the clear relationship between female sex and better eating habits [18]. This association was confirmed by a study that found a relationship between a reduction in the consumption of required macronutrients, age, and sex. This could be explained by the sociocultural conditions related to food intake [19].

Hand hygiene and movement restrictions were initially the main recommendations to prevent the spread of the virus among the population [20]; furthermore, house hygiene must be carried out daily, especially in high-contact areas, with bleach-based solutions [19]. The participants of our study reported using bleach as a main cleaning supply; however, 57.6% reported daily cleaning. On the other hand, 95% reported regular hand washing, fulfilling this recommendation, in accordance with the initial sanitary measures affecting contact exposure to the SAR-COV-2 virus, but not aerosol exposure, such as mandatory use of masks [3].

Social isolation is considered to be the best measure to reduce infection and death in a population [20], even more so when it is prematurely applied [21]. However, it was found that voluntary isolation predominated over compulsory isolation, with 50.5% and 42.6%, respectively, among the respondents. This reflects that the countries analyzed took different measures at the beginning of the pandemic.

A sedentary lifestyle and low levels of physical activity have been related to a higher risk of severe forms and higher mortality from COVID-19 [16]. In our sample, it was found that 58% of respondents did not perform physical activity during social isolation time; this coincides with the reduction in physical activity found in different contexts, although groups of people who discreetly increased their physical activity level have been identified [6]. This situation is relevant, as physical activity at the recommended levels not only promotes health, but also prevents illnesses and represents an efficient intervention in the sanitary field. It is worth mentioning that this measure is not limited to the risk prevention of transmissible illnesses, such as SARS-CoV-2, but is also applied to non-transmissible pre-existing illnesses [22].

It has been reported that an effective physical activity is one that surpasses 150 h of moderate intensity to positively affect health [23]. In this sense, our study found that only 58.8% of respondents reported accomplishing this intensity, with a mean of 43.29 ± 28.699 min; thus, this activity would be insufficient in providing health benefits. Moreover, if the quantity of necessary exercise days considered for an adequate cardiovascular health, described as an average of three days, is considered, we found that only 31.6% of participants reported exercising less [24].

With respect to pain, it was found that 54.4% of participants reported pain during physical activity, especially those with less experience [25]. In addition, one study reported late muscular pain located in the neck and spine [26], which coincides with the results of our study; thus, this area represents one of the main reported pain locations. On this matter, it is reasonable to relate pain in the spine to an increase in sedentary lifestyle [27], which was influenced by the sanitary measures of social confinement.

As mentioned earlier, isolation was initially the most useful measure for controlling the pandemic; however, isolation is associated with lifestyle and physical activity changes that often result in a sedentary lifestyle and, consequently, in an increase in cardiovascular illnesses [28] and mental health status alterations [29]. Based on the knowledge obtained during the influenza pandemic in 2009, it is known that physical activity increases early immune activation during infection [30]; for this reason, keeping up exercise habits during a pandemic is a good strategy for overall health.

A correlation between the type of isolation and the frequency of exercise was also found. In this study, it was shown that the highest proportion of people who exercised were those who lived in voluntary isolation, followed by those who lived in compulsory isolation. As far as we know, other analytical studies relating exercise habits to the type of isolation during the pandemic have not been carried out. However, it is possible to compare our results to a population with restricted mobilization, for example, using incarcerated individuals as a reference, who are deprived of liberty in prison but who reach high adherence results through the promotion of a structured exercise plan [31]. This implies a general improvement in health status. With respect to this, it is important to recommend the fulfillment of a healthy level of physical activity during social confinement.

Based on the results discussed above, physical activity is important for adults aged 18 to 64 years, who would benefit from at least 150 h of moderately intensive activity per week [23]. Supporting this idea, the World Health Assembly established goals for a 25% reduction in premature deaths by non-transmissible illnesses and reduction of 15% in physical inactivity by 2025 [32]. Considering the previous context, the existence of a significant association between the fulfillment of physical activity and age was shown; however, there was no significant relationship found between exercise intensity and age. Similarly, in a study carried out on elderly people, an association between the fulfillment of physical activity and its health benefits and age [33] was found. It is relevant to mention that the fulfillment of physical activity related to adequate nutrition is one of the most useful strategies to counteract sarcopenia and prevent disability during aging [34]. In this regard, age has been considered a relevant risk factor for COVID-19 mortality [35], in the South American case it has a lower proportion of older adults than European countries, so it has a smaller vulnerable group [4].

In addition, frequent physical activity can particularly be beneficial for those with higher disability risk [36]. For this reason, the implementation of training plans for elderly people can be important during a pandemic.

On the other hand, an association between occupation and physical activity during quarantine was found. In previous studies, it has been noted that occupation and workplace support are not associated with an increase in physical activity [37], and high levels of overweight and inactive health care workers have been found [38]. One study carried out on elderly people in Cuba showed an association between the type of previous occupation and their perception of health and psychological benefits [33].

With studies having found that people partook in physical activity more often during the quarantine period [6], it is important to mention that progressive implementation becomes vital for the adequate practice of constant exercise, as symptoms indicating a musculoskeletal lesion may be reported [39]. These data are supported by an online survey, which revealed that 12% of respondents reported having a lesion during confinement due to home exercise [40]. Following the same line, during this quarantine, the use of home workout equipment and moderately intensive exercises were recommended specially for elderly people [41].

Among the respondents, there was a relation between the intensity of physical activity performed and the pain experienced during it; on the other hand, no relation was found between the duration of physical activity and pain. It is known that a health evaluation before starting a physical exercise routine is an important strategy to prevent pain and lesions [42].

The nine Spanish-speaking countries of South America constitute a population of more than 215 million inhabitants, the demographic transition towards aging has generated an epidemiological transition to chronic diseases that have their origin in unhealthy lifestyles such as alcohol consumption, smoking, poor eating habits and sedentary lifestyles, which can be explained by different disciplines such as education, psychology, anthropology, sociology, and health sciences [43]. The main findings found in the first phase of the pandemic indicate that unhealthy lifestyles in terms of diet and physical activity were present from the beginning of the pandemic. In general, the health systems of the countries involved did not give importance to this phenomenon, despite being risk factors for severe forms of COVID-19. Therefore, future research should be aimed at investigating which factors are associated with these unhealthy lifestyles in pandemic and post-pandemic contexts. Additionally, future studies should analyze the strengths and weaknesses of countries’ health systems in the face of negative lifestyle changes during new outbreaks or future pandemics.

### Strengths and Limitations

The main goal of this study was reached; it was found that prevention measures during the pandemic—more specifically, confinement and social isolation—affected the lifestyles of South American people and caused possible long-term consequences that were potentially avoidable and preventable, such as chronic diseases (diabetes, hypertension hypertension, obesity, hyperlipidemia, others) that are highly susceptible to sedentary lifestyles and high calorie intake.

This study has some limitations. The first is a potential bias as the sample was not random, which could reduce its external validity. In order to reduce this bias, a sample larger than the estimated sample was collected, Although not all the samples by country represented the same proportion with respect to their population. In addition, the mechanisms of identification of the participants were carried out through social media; thus, populations that do not use these services or have a different native language than Spanish may not be represented. It is also possible to identify a possible measurement bias, since the questionnaires may have had a memory bias and, since they were not adapted to the context of each country, some questions may not have been understood by the participants. At the design level, one limitation identified is that a cross-sectional study cannot verify changes over time, establish causality between the variables studied.

## 5. Conclusions

This research demonstrates the necessity for lifestyle improvements, especially with regard to modifiable lifestyles such as physical activity and diet, as we found that six out of ten individuals did not perform physical activity during COVID-19 isolation. Moreover, only one out of four respondents reported having a sufficient and balanced diet. A statistically significant relationship was also found between the time (hours) spent sitting or lying down and the type of isolation and between the duration (minutes) of physical activity and type of isolation. Regarding hygiene, seven out of ten respondents reported taking daily showers and the majority indicated that they washed their hands often. 

Based on these results, it is recommended that South American governments reorient public services to promote better lifestyles, as well as consider complementary measures to those of social isolation in cases of new outbreaks, new diseases or emerging and reemerging diseases, due to the consequences they may have on the health and well-being of the population.

## Figures and Tables

**Figure 1 ijerph-19-15318-f001:**
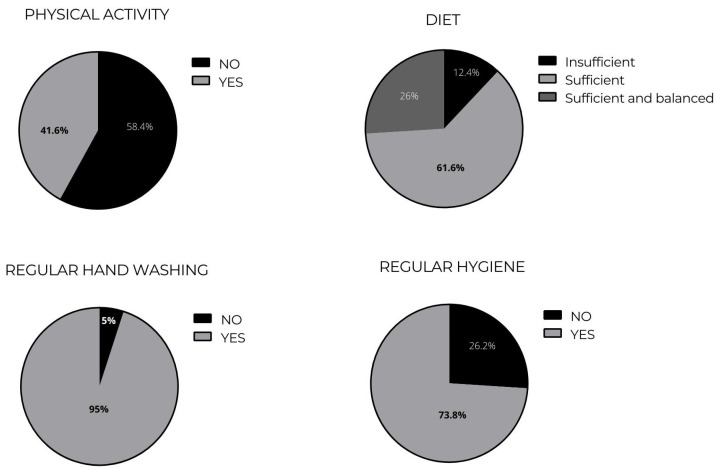
Lifestyles of adults from Spanish-speaking countries in South America during COVID-19.

**Table 1 ijerph-19-15318-t001:** Country of residence and age of respondents in the study.

Country of Residence	18 to 28		29 to 39		40 to 50		51 to 61		62 or Older		Total	
	*n*	%	*n*	%	*n*	%	*n*	%	*n*	%	*n*	%
Peru	1687	10.04	816	4.85	443	2.64	233	1.39	64	0.38	3243	19.3
Colombia	1379	8.20	565	3.36	297	1.77	158	0.94	51	0.30	2450	14.6
Argentina	1217	7.24	579	3.44	265	1.58	155	0.92	36	0.21	2252	13.4
Venezuela	1009	6.00	445	2.65	271	1.61	133	0.79	46	0.27	1904	11.3
Chile	972	5.78	381	2.27	260	1.55	177	1.05	49	0.29	1839	10.9
Paraguay	856	5.09	337	2.00	186	1.11	91	0.54	29	0.17	1499	8.9
Bolivia	854	5.08	314	1.87	154	0.92	81	0.48	25	0.15	1428	8.5
Uruguay	638	3.80	247	1.47	133	0.79	67	0.40	4	0.02	1089	6.5
Ecuador	584	3.47	215	1.28	117	0.70	63	0.37	16	0.10	995	5.9
Other	70	0.42	23	0.14	13	0.08	3	0.02	3	0.02	112	0.7
Total	9266	55.12	3922	23.33	2139	12.72	1161	6.91	323	1.92	16,811	100.0

**Table 2 ijerph-19-15318-t002:** Hygiene, mobilization, and diet variables during quarantine among study participants.

Variables		*n* = 16,811	
		*n*	%
Do you take a shower every day?	Yes	12,407	73.80
	No	4404	26.20
Do you wash your hands frequently?	Yes	15,971	95.00
	No	840	5.00
Do you go out of your residence location weekly?	Yes	10,339	61.50
	No	6472	38.50
Do you consider your diet is	Insufficient	2082	12.38
	Sufficient	10,366	61.66
	Sufficient and balanced	4363	25.95
Do you do some physical activity during the quarantine?	Yes	6996	41.62
	No	9815	58.38

**Table 3 ijerph-19-15318-t003:** Type of isolation according to physical activity and intensity level.

			Type of Isolation			Total	
			Voluntary *n* = 4030	Restricted *n* = 523	Compulsory *n* = 3488		*p*-Value
Do you do physical activity during the quarantine?	Yes	N°	3472	460	3064	6996	0.021 *
		%	20.7%	2.7%	18.2%	41.6%	
	No	N°	5024	695	4093	9812	
		%	29.9%	4.1%	24.4%	58.4%	
Intensity level of the physical activity you do	Low	N°	1726	216	1370	3312	0.008 *
		%	21.5%	2.7%	17.0%	41.2%	
	Moderate	N°	2134	290	1939	4363	
		%	26.5%	3.6%	24.1%	54.3%	
	High	N°	170	17	179	366	
		%	2.1%	0.2%	2.2%	4.6%	

* Statistically significant differences (95% reliable according to Chi-square test).

**Table 4 ijerph-19-15318-t004:** Comparison between the type of social isolation, the time (hours) spent sitting or lying down, and time (minutes) spent doing physical activity.

	Type of Isolation	*n*	Mean	Standard Deviation	Mean of Standard Error	*p*-Value
Time (hours) sitting or lying	Voluntary	8156	7.28	5.107	0.057	<0.001 *
	Other	7996	7.93	5.332	0.060	
Time (minutes) of physical activity	Voluntary	3850	42.13	27.313	0.440	<0.001 *
	Other	3876	44.43	29.972	0.481	

‘Other’ represents those who reported restricted and compulsory isolation. * Indicates statistically significant differences between independent samples with 95% reliability according to a Student’s *t*-test.

## Data Availability

By means of a reasonable request the data used in the present study can be provided by the corresponding author.

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
