# Peer review of "Lifestyles during the First Wave of COVID-19: A Cross-Sectional Study of 16,811 Adults from Spanish-Speaking Countries in South America"

_ijerph, 2022, doi:10.3390/ijerph192215318_

Round 1

Reviewer 1 Report

Dear authors,

The chosen topic is useful in the current (post) pandemic context. Research results are clearly presented. I believe that the article needs some changes and additions, which would increase its value.

In this regard,

1. (Lines 18-32). I suggest you remove the percentages from the Abstract and replace them with formulations like "the majority", "almost half", "only a small part", etc. A better representation here of the Discussion and Conclusions of the article would be useful.

2. (Lines 72-76). State your research objectives more clearly and in more detail. What lifestyle dimensions are you interested in?

3. Explain in the Introduction why you chose only Spanish speakers as subjects and correlate this option in the Discussion (research limitations) with information about the share of Spanish speakers in the South American countries you researched.

4. In the Discussion chapter, develop the data interpretation part. For example, what is the significance of the fact that only 26% of respondents believe that they had a balanced diet during the quarantine period? (Lines 224-225) Is there research on this aspect before or after quarantine? Or, what is the significance of the fact that only 57.6% (which to me personally seems like a high percentage) clean their houses daily? (Lines 244-245) These are just examples, there are many more meanings that should be highlighted, thus giving depth to the article.

5. If you talk in the Introduction about the differences from Europe (Lines 69-71), you should highlight these differences better in the Discussion

6. Better highlight the possible long-term consequences of social isolation that you refer to as being reported by research (Lines 328-329)

7. In the Conclusions I suggest you to correlate the references to the need to improve the lifestyle and the recommendations with the particularities of health, diet and appetite for physical activities of the Spanish-speaking population of South America.

Please consider all these requests and suggestions as constructive. They were formulated to improve the article. Success!

Reviewer 2 Report

Very well prepared article. Survey conducted on a large sample of respondents.

In my opinion, there is no need to highlight so many subsections in Chapter 2. Study Design and Data Analysis is enough.

Information on age groups and country of origin of respondents - table 1 and its description I would move to chapter 2 - characteristics of the research sample.

Please provide the percentages in the graphs (figure 1).

Line 194-204. a lot of statistics were presented but no details of groups (% of responses) were given as in Table 3 and 4.

The discussion refers to the most important results. Good selection of literature. Current literature from recent years. 

Reviewer 4 Report

Dear Authors

This cross-sectional manuscript is eminently descriptive.

The authors must explain the importance of knowing the changes in lifestyles due to a state of emergency. The authors must inform more about the instrument used, what type of validation it obtained before its application, and how the questions were adjusted according to the different idioms of the Spanish language of each country. The responses may have been biased by language.

Lines 70-71. Please explain what sociocultural conditions in Latin America compared to Europe make it less vulnerable to COVID, not just age.

Lines 72-76: This paragraph is not clear. Please mention why the study is important and what it contributes to the literature. Why is it important to compare populations...

I don't see much point in the discussion.

The authors compare their results against other studies under different conditions, which does not make sense.

The harmful effects of confinement, a sedentary lifestyle, and an inadequate diet on people's health are already known. The manuscript constantly repeats these effects.

Personally, it would make better sense to study the strengths and weaknesses of the health systems of the countries in the face of the negative changes in lifestyles during the pandemic.

Lines 235-240: This discussion makes no sense.

Lines 250-251: This explanation doesn't make sense.

It is difficult to know the pandemic's effects on the population if it is only studied once. These types of studies should include retrospective analyses; that is, the items should also ask about pre-pandemic lifestyles.

According to the present results and those of the literature, please add a conceptual map or an illustrative figure that summarizes all the changes observed in the population's lifestyles during the pandemic. According to this figure, discuss the changes.

Recommendations have nothing to do with results. A possible problem to address is the inability of states and institutions to deal with the pandemic, where to date, the necessary changes have not been made to better prevent or deal with another natural disaster. The rest is already covered.

Round 2

Reviewer 1 Report

Dear authors,

If you want to keep a few percentages in the Abstract, please make sure that they are the most relevant research results.

Please check the wording of the newly inserted passages, there are places where punctuation marks are missing and places where they are doubled.

Success!

Reviewer 4 Report

Dear authors

It is a pleasure to see how a manuscript is transformed to be suitable for publication. Congratulations on improving the introduction and discussion. I have no further comments.

Author Response

Dear Reviewer.
We appreciate your contributions for the substantial improvement of the manuscript. Finally, the general wording of the article has been revised and all the references have been revised. In addition, we have relied on a native editor to standardize the writing style.

Thank you very much